# Perinatal Ethanol Exposure Induces Astrogliosis and Decreases GRP55/PEA-Mediated Neuroprotection in Hippocampal Astrocytes of the 3×Tg Alzheimer’s Animal Model

**DOI:** 10.3390/ijms262211154

**Published:** 2025-11-18

**Authors:** Miguel Rodríguez-Pozo, Beatriz Pacheco-Sánchez, Meriem Ben Rabaa, Marialuisa de Ceglia, Sonia Melgar-Locatelli, Ignacio Santos, Fernando Rodríguez de Fonseca, Juan Suárez, Patricia Rivera

**Affiliations:** 1Instituto de Investigación Biomédica de Málaga y Plataforma en Nanomedicina—IBIMA Plataforma BIONAND, 29590 Málaga, Spain; rpmiguel@uma.es (M.R.-P.); beatriz.pacheco@ibima.eu (B.P.-S.); meriem.benrabaa@outlook.com (M.B.R.); marialuisa.deceglia@ibima.eu (M.d.C.); 0619587398@uma.es (S.M.-L.); fernando.rodriguez@ibima.eu (F.R.d.F.); 2Departamento de Anatomía Humana, Medicina Legal e Historia de la Ciencia, Universidad de Málaga, 29010 Málaga, Spain; isantos@uma.es; 3Unidad de Gestión Clínica de Neurología, Hospital Universitario Regional de Málaga, 29010 Málaga, Spain; 4Molecular Biotechnology, FH Campus Wien, University for Applied Sciences, Favoritenstraße 222, 1100 Vienna, Austria; 5Unidad de Gestión Clínica de Salud Mental, Hospital Universitario Regional de Málaga, 29010 Málaga, Spain; 6Departamento de Psicobiología y Metodología de las Ciencias del Comportamiento, Universidad de Málaga, 29010 Málaga, Spain; 7Departamento de Nutrición y Bromatología, Universidad de Granada, Campus Universitario de Cartuja, 18071 Granada, Spain

**Keywords:** Alzheimer’s disease, astrocytes, hippocampus, inflammation, endocannabinoid system, GPR55, toxicology, legal medicine

## Abstract

Prenatal ethanol exposure (PEE) alters fetal brain development, potentially increasing the risk of neurodegenerative diseases such as Alzheimer’s disease (AD) later in life. Although glial activation is implicated in AD pathology via cannabinoid and neuroinflammatory signaling, its potential response to PEE in the developing brain and its contribution to AD pathogenesis remain unknown. Using 3×Tg-AD offspring of both sexes born to mothers with PEE, we analyzed astrogliosis, inflammatory markers, and key components of cannabinoid and Ca^2+^ signaling in primary cultures of hippocampal astrocytes, elements whose dysfunction contributes to neurodegeneration. Our results indicated that PEE increased astrogliosis/inflammatory response (significant elevation of *Gfap* and *Tnfα* expression) in hippocampal astrocytes at birth. This neuroinflammation was significantly associated with lower expression of cannabinoid receptors (*Cnr1* and *Gpr55*), and decreased concentrations of the anti-inflammatory lipid PEA in the culture medium, probably due to a deregulated endocannabinoid enzymatic machinery (NAPE-PLD/FAAH ratio). This research provides insights into GRP55/PEA-mediated signaling as a potential hippocampal astrocytic mechanism influenced by maternal ethanol exposure, which may contribute to neurobiological changes associated with increased vulnerability to AD-related pathology.

## 1. Introduction

Prenatal ethanol exposure (PEE) is a recognized teratogen associated with a broad spectrum of neurodevelopmental and structural anomalies, collectively termed fetal alcohol spectrum disorders (FASD). These disorders are characterized by persistent cognitive and behavioral impairments, with particularly pronounced effects on the central nervous system (CNS) [1,2,3,4,5]. Ethanol disrupts key neurobiological processes, including neural progenitor proliferation, differentiation, and migration, ultimately contributing to neurodegeneration [6]. In parallel, PEE induces long-lasting epigenetic modifications, such as altered histone deacetylase activity and microRNA expression, which shape postnatal cellular phenotypes in both in vivo and in vitro models [1,2,3,4].

Astrocytes, a significant class of glial cells, play a crucial role in CNS homeostasis, providing metabolic support, regulating synaptic transmission, and modulating neuroinflammatory responses. Astrocyte proliferation and differentiation is impaired by PEE during critical neurodevelopmental windows, resulting in aberrant expression of glial fibrillary acidic protein (GFAP) and other markers essential for astrocytic function [7]. These alterations may contribute to the neurodevelopmental deficits observed in FASD, including impaired synaptic plasticity and dysregulated neuroinflammation.

A pivotal signaling system in astrocytic function is the endocannabinoid system (ECS), which modulates neuroinflammation, neuroprotection, and glial-neuron communication [8]. The ECS comprises cannabinoid receptors, their endogenous lipid ligands, collectively termed endocannabinoids (e.g., N-arachidonoylethanolamine [AEA], also known as anandamide, and 2-arachidonoylglycerol [2-AG]), and the enzymes responsible for their biosynthesis and degradation [9,10]. Endocannabinoids are synthesized on demand in response to stimuli such as elevated intracellular calcium, metabolic stress, or cellular injury [11]. These ligands primarily engage cannabinoid receptor type 1 (CB1/CNR1) and type 2 (CB2/CNR2), modulating neuronal and glial activity. In addition to canonical endocannabinoids, paracannabinoids, such as saturated and unsaturated acylethanolamides (e.g., palmitoylethanolamide [PEA], oleoylethanolamide [OEA], and stearoylethanolamide [SEA]), exert their effects through alternative targets, including transient receptor potential vanilloid type 1 (TRPV1), G protein-coupled receptor 55 (GPR55), and peroxisome proliferator-activated receptors (PPARs), thereby expanding the functional role of ECS-mediated signaling in astrocytes [11,12].

Recent studies have implicated ECS dysregulation in the pathogenesis of Alzheimer’s disease (AD), with several ECS components emerging as potential therapeutic targets [13,14]. Evidence supports the neuroprotective effects of PEA in both in vitro and in vivo models of AD [15]. Our group previously demonstrated that glial-specific cannabinoid signaling plays a critical role in AD-related neurobiology. In 3×Tg-AD mouse models, which harbor mutations in hAPP, PSEN1, and tau genes, hippocampal astrocytes exhibit intrinsic ECS alterations and a constitutive attenuation of astrocyte activity, potentially contributing to disease progression and impaired neuroimmune regulation [16].

Alcohol consumption has also been identified as a potentially modifiable risk factor in AD, though its impact on disease onset and progression remains complex and context-dependent [17]. Variables such as dosage, drinking patterns, genetic susceptibility, nutrition status, and the duration of exposure may influence neuroinflammatory responses, oxidative stress, and synaptic integrity [18,19]. For instance, increased 2-AG production in response to Aβ-induced neurodegeneration may represent a compensatory mechanism, whereas chronic ethanol intake that suppresses 2-AG levels can exacerbate AD pathology [18]. Moreover, CB2 receptors are overexpressed in Aβ plaque-associated astrocytes and microglia in AD patients, and their stimulation mitigates Aβ-induced neurodegeneration, improving cognitive outcomes in rodent models [20].

Despite growing evidence linking PEE, ECS dysregulation, and AD pathology, the long-term consequences of early-life ethanol exposure on neuroinflammation and endocannabinoid signaling remain poorly understood. In particular, the potential priming effects of PEE on astrocytic homeostatic systems and their contribution to AD susceptibility have not been systematically investigated [21].

We hypothesize that PEE alters endocannabinoid and inflammatory signaling in astrocytes, thereby increasing vulnerability to neurodegenerative diseases later in life. To test this, we used primary cultures of hippocampal astrocytes derived from 3×Tg-AD mice to identify sex-specific astrocytic mechanisms disrupted by maternal ethanol exposure in the context of AD pathology.

## 2. Results

### 2.1. mRNA Expression of Inflammatory/Astrogliosis Factors

To assess whether hippocampal astrocytes from 3×Tg-AD pups born to mothers with PEE exhibit increased inflammatory reactivity or astrogliosis, we analyzed the gene expression of inflammatory markers and markers indicative of glial reactivity: glial fibrillary acidic protein (*Gfap*), vimentin, tumor necrosis factor α (*Tnfα*), interleukins (*Il1β* and *Il6*), and prostaglandin-endoperoxide synthase 2 (*Ptgs* or *Cox2*).

A PEE effect on *Gfap* mRNA expression was found [F (1,17) = 7.911; *p* < 0.05]. An interaction between the two factors analyzed was also found in the *Gfap* expression [F (1,17) = 21.10; *p* < 0.001]. Hippocampal astrocytes from ethanol 3×Tg-AD male offspring had higher *Gfap* mRNA levels than control 3×Tg-AD male and ethanol 3×Tg-AD female offspring (Tukey’s test; */*** *p* < 0.05/0.01; Figure 1A). Two-way ANOVA indicated effects of sex and PEE on *Tnfα* mRNA levels [F (1,20) = 4.666, *p* < 0.05; F (1,20) = 8.535, *p* < 0.01; respectively], with an overall increase in *Tnfα* mRNA levels in hippocampal astrocytes from ethanol 3×Tg-AD offspring of both sexes compared to respective control astrocytes (Figure 1C). No main effects on *Vimentin*, *Il6*, *Il1b,* and *Ptgs* mRNA levels were found (Figure 1B,D–F).

### 2.2. mRNA Expression of Cannabinoid-Related Receptors

Two-way ANOVA indicated a PEE effect and an interaction between sex and PEE [F (1,29) = 5.632, *p* < 0.05; F (1,29) = 6.283, *p* < 0.05; respectively] on *Cnr1* mRNA levels, with hippocampal astrocytes from ethanol 3×Tg-AD male offspring showing lower *Cnr1* levels than control 3×Tg-AD male offspring (Tukey’s test; * *p* < 0.05; Figure 2A). A PEE effect on *Gpr55* mRNA expression was found [F (1,28) = 11.81; *p* < 0.01]. Hippocampal astrocytes from ethanol 3×Tg-AD (male and female) offspring had lower *Gpr55* mRNA levels than control astrocytes (Figure 2C). We also observed a sex effect on *Pparα* mRNA expression [F (1,20) = 14.41; *p* < 0.01], with an overall decrease in *Pparα* mRNA levels in female hippocampal astrocytes compared to those from males (Figure 2D). No main effects on the mRNA levels of *Cnr2* and *Trpa1* were observed (Figure 2B,E).

### 2.3. mRNA Expression of Enzymes Responsible for Endocannabinoid Synthesis and Degradation

Regarding the enzymatic machinery for endocannabinoid biosynthesis and degradation, an interaction between sex and PEE was found on *Daglβ* mRNA levels [F (1,31) = 11.52; *p* < 0.01], with hippocampal astrocytes from ethanol 3×Tg-AD male offspring showing lower *Daglβ* levels than control 3×Tg-AD male offspring. We also observed lower *Daglβ* mRNA levels in hippocampal astrocytes from control 3×Tg-AD female offspring than control 3×Tg-AD male offspring (Tukey’s test; */** *p* < 0.05/0.01; Figure 3B). Two-way ANOVA also showed a PEE effect on *Mgll* and *Faah* mRNA levels [F (1,28) = 16.17, *p* < 0.001; F (1,27) = 24.56, *p* < 0.001; respectively], with a general decrease in the gene expression of these enzymes in the hippocampal astrocytes of ethanol 3×Tg-AD offspring of both sexes compared to their controls (Figure 3C,E). No main effects on the mRNA levels of *Dagl*α and *Nape-pld* were found (Figure 3A,D).

To evaluate the equilibrium between the production and degradation of endocannabinoids, we determined the ratio of the enzymes involved in the synthesis and degradation of acylglycerols and NAEs. Two-way ANOVA showed an interaction between sex and PEE on the *Daglβ*/*Mgll* ratio [F (1,31) = 11.52; *p* < 0.01]. Tukey post hoc analysis indicated a decreased *Daglβ*/*Mgll* ratio in the hippocampal astrocytes from ethanol 3×Tg-AD males compared to control 3×Tg-AD males. We also observed a lower *Daglβ*/*Mgll* ratio in the hippocampal astrocytes from control 3×Tg-AD females than control 3×Tg-AD males (Tukey’s test; */** *p* < 0.05/0.01; Figure 3G). Effects of sex [F (1,28) = 4.810; *p* < 0.05] and PEE [F (1,28) = 17.39; *p* < 0.001], as well as an interaction between factors [F (1,28) = 7.870; *p* < 0.01], were found on the *Nape-pld*/*Faah* ratio, with a significant increase in this ratio in the hippocampal astrocytes of ethanol 3×Tg-AD male offspring compared to control male offspring and ethanol 3×Tg-AD female offspring (Tukey’s test; **/*** *p* < 0.01/0.001; Figure 3G).

### 2.4. Protein Expression of Cannabinoid-Related Receptors

Two-way ANOVA indicated an interaction between sex and PEE on CB2 protein expression [F (1,20) = 9.820; *p* < 0.01]. Tukey post hoc analysis showed an increase in CB2 expression in the hippocampal astrocytes from ethanol 3×Tg-AD females compared to control 3×Tg-AD females (Tukey’s test: ** *p* < 0.01; Figure 4B). We also observed lower levels of CB2 in control female astrocytes than control male group (Tukey’s test: * *p* < 0.05; Figure 4B). No main effects on the CB1 protein expression were found (Figure 4A). A sex effect [F (1,20) = 8.561; *p* < 0.01] and an interaction between sex and PEE on PPARα protein expression were also observed [F (1,20) = 5.070; *p* < 0.05], with a decrease in PPARα protein levels in female control astrocytes compared to male control astrocytes (Tukey’s test; ** *p* < 0.01; Figure 4C). Finally, two-way ANOVA showed a sex effect [F (1,20) = 5.807; *p* < 0.05] on GPR55 protein expression, suggesting a decreased expression in females (Figure 4D). Representative immunoblots are shown in Figure 4E. Unedited blots are shown in Appendix A.

### 2.5. Protein Expression of Enzymes Responsible for Endocannabinoid Synthesis and Degradation

Two-way ANOVA showed a PEE effect [F (1,20) = 15.60; *p* < 0.001] and an interaction between factors on DAGLα protein expression [F (1,20) = 6.925; *p* < 0.05]. Tukey post hoc analysis indicated a decrease in DAGLα expression in the hippocampal astrocytes from ethanol 3×Tg-AD females compared to control 3×Tg-AD females (Tukey’s test: *** *p* < 0.001; Figure 5A). A PEE effect on MAGL protein expression was found [F (1,20) = 5.209; *p* < 0.05], suggesting increased MAGL protein levels in the hippocampal astrocytes from ethanol 3×Tg-AD offspring of both sexes compared to control astrocytes (Figure 5C). An interaction between PEE and sex on NAPE-PLD protein levels were also found [F (1,20) = 9.135; *p* < 0.01] (Figure 5D). No main effects on the DAGLβ and FAAH protein expression were found (Figure 5B,E).

To analyze the balance between endocannabinoid production and degradation, we calculated the ratio of the enzymes involved in the synthesis and degradation of NAEs and acylglycerols. Two-way ANOVA showed a PEE effect and an interaction between factors on the DAGLα/MAGL ratio [F (1,20) = 13.99, *p* < 0.01; F (1,20) = 5.696, *p* < 0.05; respectively]. Tukey post hoc analysis indicated a decreased DAGLα/MAGL ratio in the hippocampal astrocytes from ethanol 3×Tg-AD females compared to control 3×Tg-AD females. We also observed that hippocampal astrocytes from control 3×Tg-AD females showed a higher DAGLα/MAGL ratio than control 3×Tg-AD males (Tukey’s test: */** *p* < 0.05/0.01; Figure 5F). We also found an interaction between sex and PEE in the NAPE-PLD/FAAH ratio [F (1,20) = 5.846; *p* < 0.05] (Figure 5H). Representative immunoblots are shown in Figure 5I. Unedited blots are shown in Appendix A.

### 2.6. Culture Medium Endocannabinoid and Paracannabinoid Levels

Concentrations of the acylglycerols 2-arachidonoylglycerol (2AG) and 2-linoleoylglycerol (2LG), in addition to the NAEs palmitoleoylethanolamine (POEA), linoleoylethanolamine (LEA), palmitoylethanolamine (PEA), oleoylethanolamine (OEA), and *N*-stearoylethanolamine (SEA) were measured in the culture medium of hippocampal astrocytes.

Two-way ANOVA only indicated a PEE effect on PEA levels [F (1,14) = 5.983; *p* < 0.05], with decreased PEA levels in the hippocampal astrocytes from ethanol 3×Tg-AD offspring of both sexes compared to control astrocytes (Figure 6E). No main effects on the remaining endocannabinoid and paracannabinoid concentrations were observed (Figure 6A–D,F,G).

### 2.7. mRNA Expression of Ca^2+^ Signaling-Related Genes

Since astrocytes communicate with other brain cells through mechanisms primarily dependent on Ca^2+^, which are in turn controlled by the ECS, we analyzed the expression of genes important in astrocytic Ca^2+^ dynamics. An interaction between sex and PEE was found on the mRNA levels of the ionotropic channel *P2rx5* [F (1,20) = 14.27; *p* < 0.01]. Tukey’s test showed lower mRNA levels of *P2rx5* in the hippocampal astrocytes from control 3×Tg-AD females than control 3×Tg-AD males (** *p* < 0.01; Figure 7A).

Glutamate toxicity is implicated in the pathogenesis of AD, with astrocytes playing a key role through cannabinoid signaling. Therefore, to investigate potential glutamate toxicity, we examined the expression of two isoforms of glutaminase, the enzyme that produces glutamate from glutamine. We observed a sex-dependent effect on *Gls* mRNA expression [F (1,20) = 16.50; *p* < 0.001], with hippocampal astrocytes from control 3×Tg-AD females showing lower *Gls* mRNA levels than control 3×Tg-AD males (Figure 7E).

## 3. Discussion

Prenatal ethanol exposure (PEE) is a well-established risk factor for fetal alcohol spectrum disorders (FASD), which encompass a range of developmental impairments, particularly affecting the central nervous system (CNS). Ethanol interferes with key developmental processes, notably disrupting the function and maturation of astrocytes, which are essential glial cells that maintain neuronal homeostasis and provide synaptic support. Emerging evidence suggests that the endocannabinoid system (ECS) plays a role in modulating astrocytic function and neuroinflammatory responses. Given its role in neuroprotection, dysregulation of the ECS may represent a mechanistic link between PEE-induced astrocytic and long-term neuropathological outcomes, including those associated with Alzheimer’s disease (AD).

Chronic alcohol consumption has been shown to diminish protective levels of endocannabinoids and related lipid mediators, potentially exacerbating the progression of AD. Considering this premise, the present study investigated the impact of PEE on astrocytic activation and ECS signaling within the context of AD pathology. Utilizing the 3×Tg-AD mouse model, which harbors key genetic mutations associated with AD, this research provides novel insights into how PEE may predispose individuals to long-term neuroinflammatory changes and heightened vulnerability to neurodegenerative disorders.

Our findings revealed that PEE may elicit a pronounced inflammatory response in male hippocampal astrocytes at birth, as evidenced by elevated levels of GFAP and TNFα. This astrocytic activation was accompanied by significant downregulation of the cannabinoid receptors CB1 and GPR55, indicating impaired endocannabinoid signaling. In addition, concentrations of palmitoylethanolamide (PEA), a neuroprotective paracannabinoid, were markedly reduced in the astrocytic culture medium, further supporting a disruption in ECS-mediated homeostasis mechanisms.

Reactive astrogliosis is one of the earliest pathological hallmarks of AD in both human and rodent brain tissues [22]. In our previous research, however, we observed no significant differences in GFAP expression in the hippocampal astrocytes of neonatal 3×Tg-AD mice compared to wild-type (WT) controls, suggesting that astrocytic activation may emerge later in the disease course [16]. In contrast, the present study, which focused on 3×Tg-AD offspring born to either control or ethanol-exposed mothers, demonstrated a marked upregulation of Gfap mRNA in hippocampal astrocytes of male neonates following PEE. This sex-specific increase in Gfap expression at birth suggests that maternal ethanol intake may trigger astrocytic reactivity and neuronal damage in the hippocampus of 3×Tg-AD animals. These findings align with the growing evidence that reactive astrocytes contribute to the initiation and progression of AD pathology through inflammatory and neurotoxic mechanisms [22].

In addition to GFAP, our data revealed a significant increase in Tnfα mRNA levels associated with PEE, especially in male hippocampal astrocytes, suggesting a sex-specific proinflammatory response to ethanol-induced neurotoxicity. This finding is consistent with previous reports linking reactive astrogliosis to elevated cytokine expression and broader neuroinflammatory processes in a context of AD pathology [7]. Notably, no significant changes were observed in the expression of other inflammatory markers analyzed, suggesting a selective activation of inflammatory pathways in response to PEE. It is essential to recognize that the potential interference of in vitro culture conditions may influence the manifestation of inflammation-associated phenotypes, potentially masking or modulating the full spectrum of astrocytic responses.

This partial inflammatory response observed in hippocampal astrocytes following maternal ethanol consumption may be related to disrupted endocannabinoid signaling, particularly through astroglial CB1 receptors [23]. Endocannabinoids are known to exert anti-inflammatory effects via CB1 receptor activation, and astrocytic CB1 receptors play a pivotal role in regulating synaptic plasticity, memory formation, and behavioral outcomes—functions that are notably vulnerable to alcohol-induced perturbations [23,24,25]. Previous findings by Bonilla del Río et al. [12] demonstrated a reduction in CB1 receptor expression in astrocytes exposed to maternal ethanol intake, supporting the hypothesis that ethanol impairs astroglial anti-inflammatory responses. In our study, this downregulation was more pronounced in male offspring, reinforcing the notion of a sex-dependent vulnerability to ethanol-induced neuroinflammation.

Furthermore, previous studies in human AD brains reported that CB1 receptor expression is markedly reduced and CB2 receptor levels are elevated in correlation with amyloid-beta (Aβ) accumulation [26,27]. Our study observed a similar pattern in hippocampal astrocytes of offspring exposed to PEE. Specifically, PEE led to a downregulation of CB1 and an upregulation of CB2 expression, suggesting that maternal ethanol intake may accelerate AD-like neuropathological changes by disrupting endocannabinoid signaling. This shift in cannabinoid receptor expression may reflect a compensatory response to heightened neuroinflammation and Aβ-related pathology, further supporting the hypothesis that early-life ethanol exposure contributes to the early onset of neurodegenerative diseases.

In addition to alterations in classical cannabinoid receptors, our study reveals a significant decrease in Gpr55 mRNA expression in hippocampal astrocytes of both male and female offspring with PEE. This finding contrasts with previous reports demonstrating robust upregulation of GPR55 in the hippocampus of various AD mouse models, where its expression correlates with the stage and severity of Aβ-related pathology [28,29]. Notably, GPR55 activation has neuroprotective effects, as pharmacological stimulation of this receptor attenuates cognitive impairment, neurotoxicity, neuroinflammation, and synaptic dysfunction in AD mouse models [30,31]. Therefore, the ethanol-induced suppression of Gpr55 expression observed in our study may compromise astrocyte-mediated neuroprotection and contribute to the hippocampus’s early vulnerability to AD-related pathology.

In our preliminary study [16] comparing hippocampal astrocytes from WT versus 3×Tg-AD mice at birth, we found no significant differences in Gpr55 expression, which aligns with previous reports linking GPR55 upregulation to AD progression and Aβ plaque accumulation [28,29]. Interestingly, in the present study, we observed that PEE led to a significant decrease in Gpr55 mRNA levels in hippocampal astrocytes of 3×Tg-AD offspring at birth, potentially compromising its neuroprotective functions. We therefore hypothesize that the PEE-induced suppression of this receptor could represent a critical mechanism by which early-life ethanol exposure accelerates vulnerability to AD.

Currently, there are limited studies directly linking perinatal programming to alterations in Gpr55 expression and its role in the pathogenesis of neurological diseases. However, studies using animal models of autism spectrum disorder and related neurodevelopmental conditions have reported altered PEA signaling in response to social interactions, alongside changes in the expression and activity of key enzymes involved in the synthesis and degradation of endocannabinoids. Notably, these models also exhibited downregulation of Pparα and Gpr55 mRNA levels in the brain [32,33], suggesting that early-life environmental factors may influence cannabinoid-related signaling pathways with long-term consequences for neurodevelopment and behavior.

Although our results do not reveal significant changes in PPARα expression, we observed a marked reduction in PEA levels in the culture medium of hippocampal astrocytes from offspring exposed to prenatal ethanol, with the effect being more pronounced in males. This reduction may reflect a disruption in endogenous lipid signaling pathways involved in neuroprotection and inflammation regulation. Supporting this notion, PEA supplementation has been shown to exert beneficial effects in various animal models of autism spectrum disorder and related neurodevelopmental conditions, improving behavioral outcomes and modulating key processes such as neuroinflammation, neurotrophic support, apoptosis, neurogenesis, synaptic plasticity, and neurodegeneration [34,35]. These findings suggest that ethanol-induced depletion of PEA may compromise astrocyte-mediated neurodevelopmental mechanisms, potentially contributing to increased susceptibility to AD-related pathology.

PEA is an endogenous lipid mediator known for its potent neuroprotective and anti-inflammatory properties, primarily exerted through activation of receptors such as PPARα and GPR55 [36,37]. Studies have shown that PEA counteracts Aβ-induced brain damage and cognitive decline in AD models, with doses of 10–30 mg/kg improving memory and neuronal health [38,39]. Scuderi et al. [40] demonstrated that PEA, via activation of PPARα, protects primary rat astrocytes by mitigating reactive gliosis induced by Aβ. Supporting this finding, chronic PEA administration has been shown to reduce glial activation and suppress proinflammatory responses in rats with AD, highlighting its potential as a modulator of neuroinflammation. Moreover, pharmacological inhibition of PEA-degrading enzymes has emerged as a promising therapeutic strategy to counteract neuroinflammation [41,42]. Overall, PEA appears to be a promising therapeutic candidate for neurodegenerative diseases such as AD.

As discussed above, our results showed a significant reduction in PEA levels in hippocampal astrocytes following PEE, accompanied by a downregulation of GPR55 expression, while PPARα levels remained unchanged. This selective disruption of the GPR55/PEA signaling suggests that maternal ethanol intake may impair astrocyte-mediated neuroprotective mechanisms critical for mitigating early neuroinflammatory responses and maintaining neuronal integrity. To support the hypothesis that PEE compromises neuroprotection and may accelerate the onset or progression of AD-related pathology via the GPR55/PEA pathway, further studies focusing on long-term consequences are needed.

Concerning the decreased concentration of PEA observed in the hippocampal astrocytes of 3×Tg-AD offspring following PEE, our results revealed a marked downregulation of fatty acid amide hydrolase (Faah), the primary enzyme responsible for PEA degradation. Concurrently, we observed an increased Nape-pld/Faah ratio, suggesting a shift in the balance between synthesis and degradation of NAEs. This alteration may reflect a compensatory response aimed at preserving PEA levels in the face of ethanol-induced depletion. Previous studies have shown that modulation of FAAH activity can influence endocannabinoid tone and neuroinflammatory outcomes, with reduced FAAH expression potentially serving as a protective mechanism against neurodegeneration [41,42]. Thus, the observed enzymatic changes may represent an adaptive attempt by astrocytes to restore homeostasis and mitigate the loss of PEA-mediated neuroprotection in the context of early-life ethanol exposure.

Endocannabinoids are lipid-based neuromodulators synthesized on demand and released from cells in response to elevated intracellular calcium levels, allowing for rapid and localized signaling within the CNS. Concerning AD, endocannabinoids have been shown to exert neuroprotective effects by attenuating glutamate excitotoxicity, primarily through the inhibition of presynaptic glutamate release [43,44]. Astrocyte-cannabinoid signaling plays a significant role in this process, as it contributes to the modulation of synaptic activity, neuroinflammation, and neuronal survival. Disruption of this astrocyte-cannabinoid axis—such as that induced by PEE—may therefore compromise the brain’s ability to counteract excitotoxic damage and accelerate neurodegenerative processes associated with AD.

In our previous study [18], we reported that female astrocytes from 3×Tg-AD mice exhibited reduced subcellular Ca^2+^ transit, regulated by CB1 receptors, alongside a reduction in the expression of calcium-handling proteins, including the mitochondrial calcium uniporter (Mcu) and inositol 1,4,5-trisphosphate receptor type 1 (Itpr1). However, in the present study, PEE did not significantly affect Ca^2+^ signaling or glutamate-mediated excitotoxicity in astrocytes, indicating that these pathways may not be central to the astrocytic response to PEE in the context of AD. Instead, we observed reduced expression of glutaminase and NMDA receptor subunits in hippocampal astrocytes of 3×Tg-AD mice, particularly in females, suggesting a diminished capacity for glutamate synthesis and release. These results indicate that astrocytic glutamate excitotoxicity may play a limited role in the early development of AD pathology, at least under conditions of PEE, and highlight the importance of alternative mechanisms, such as neuroinflammation and endocannabinoid signaling, in disease progression [43,44].

Notably, our findings reveal sex-specific alterations in the expression of purinergic receptor P2rx5, an ATP-gated ion channel essential for calcium signaling and astrocytic reactivity. P2rx5 has been implicated in modulating neuroinflammatory responses and glial activation, processes that are increasingly recognized as contributors to AD. Similarly, alterations in glutaminase expression (Gls) suggest potential disruption in glutamate homeostasis, a key factor in excitotoxicity and synaptic dysfunction associated with AD. These sex-specific differences in astrocytic gene expression could be associated with the different vulnerability and progression of AD pathology observed in males and females, reinforcing the importance of considering sex as a biological variable in neurodegenerative research.

Taken together, the pronounced sex-dependent differences observed in astrocytic signaling, particularly the heightened inflammatory response (elevated GFAP and TNFα) and greater dysregulation of the endocannabinoid system (CB1, GPR55, and PEA) in male offspring at birth, represent a critical finding with potential epidemiological relevance. This differential vulnerability may reflect a molecular basis for the sex biases documented in both FASD and AD. Studies show that males are often disproportionately affected by the severity of FASD-related neurodevelopmental and cognitive deficits [45]. Our results suggest that PEE may program a more vulnerable male brain phenotype from birth. Although AD prevalence is higher in women, sex differences in disease progression, pathology, and response to neuroinflammatory insults have been consistently reported. The PEE-induced early disruption of astrocytic function and loss of neuroprotective signaling might impact the timing and manifestation of AD pathology later in life in a sex-dependent manner. Thus, these findings support the hypothesis that perinatal insults, acting through sex-specific astrocytic signaling, may contribute to differential susceptibility, onset, and progression of both neurodevelopmental and neurodegenerative disorders.

Limitations: While the present study offers valuable insights into the direct cellular and molecular effects of PEE on hippocampal astrocytes in the 3×Tg-AD mouse model, its findings must be interpreted within the inherent limitations of an in vitro system. The use of primary astrocytic cultures, although powerful for dissecting cell-specific responses, inherently removes astrocytes from the complex and dynamic brain microenvironment. This isolation excludes critical interactions and signaling with neighboring neurons, microglia, endothelial cells, and extracellular matrix components, all of which contribute to the integrated neuroinflammatory and neuroprotective mechanisms observed in vivo. Consequently, artificial culture conditions may not fully replicate the spatial and temporal dynamics, as well as the coordinated cell-to-cell signaling pathways, of astrocyte behavior in the intact brain. Moreover, while our results suggest that PEE induces early astrocytic dysregulation and sex-specific vulnerability, particularly in males, these molecular changes require validation in vivo to determine their relevance to behavioral outcomes and disease progression. Future studies using PEE-exposed 3xTg-AD mice in longitudinal designs are essential to confirm these findings, assess functional consequences, and elucidate the contribution of astrocytic alterations to the pathogenesis of both FASD and AD.

## 4. Materials and Methods

### 4.1. Ethics Statement

All procedures were conducted in strict adherence to the principles of laboratory animal care (National Research Council, Neuroscience CoGftUoAi, Research B, 2003) following the European Community Council Directive (86/609/EEC) and were approved by the Ethical Committee of the University of Málaga (09/06/2021/095). Special care was taken to minimize the suffering and the number of animals needed to perform the procedures.

### 4.2. Animal Model

The 3×Tg-AD mouse model (Jackson Laboratory, Bar Harbor, ME, USA), which combines mutant hAPP (Swedish), PSEN1 (MM146V), and tau (P301L) transgenes and results in Aβ and tau pathologies, was used in the present study. Adolescent (3–4 weeks old) female 3×Tg-AD mice were individually housed in standard cages along experiments (pre-gestation and gestation), and maintained under controlled room conditions: 21 ± 1 °C room temperature, 40 ± 5% relative humidity, and a 12 h light-dark cycle (lights off at 20:00 h).

### 4.3. Experimental Procedures on Animals

Animal experimentation was carried out according to the schedule described in Figure 8A. Three weeks before mating (pregestational period), animals were handled, weighed (17–18 g), and randomly assigned to a control or PEE. During the pregestational period, ethanol was administered once a week at a concentration of 4 g/kg orally (gavage).

After the pregestational period, female mice were allowed to mate with sexually mature adult male mice (aged 8 weeks) of the same strain in their home cage for 72 h. The presence of spermatozoa plugs in the vaginal smear confirmed successful mating, which was designated as gestational day 0 (GD0). During gestation, dams were maintained on control or PEE as in the pregestational period. Dams were exposed ad libitum to increasing amounts of ethanol, from 2.5 to 10% (*v*/*v*), dissolve in the tap water bottle. The solution was the sole fluid source in the cage. Volume consumption was daily monitored. A daily average of 0.85 g of pure ethanol was finally consumed during pregnancy, once 10% is reached. Of the total number of litters obtained at birth (a litter per dam), 5 litters per condition (control and PEE) were randomly selected, and a total of 20 pups (10 pups per group, 5 males and 5 females each) were sacrificed by decapitation on postnatal day (PND) 2–3.

### 4.4. Primary Cultures of Hippocampal Astrocytes

In vitro experimentation was carried out according to the schedule described in Figure 8B. On PND2–3, primary astrocytes were cultured from a pool of hippocampal cells obtained from 5 male and 5 female pups, separately, that were born to control and PEE mothers, as described previously [16,46].

On PND2–3, pups were decapitated, and their hippocampi were removed under sterile conditions using a magnifying glass and triturated in DMEM F-12 (Gibco, Grand Island, NY, USA) containing 1% penicillin-streptomycin (Gibco, Grand Island, NY, USA). The suspension was centrifuged, and the pellet was resuspended in DMEM: F12 (Gibco, Grand Island, NY, USA) supplemented with 10% fetal bovine serum (FBS) and 1% antibiotic-antimycotic solution. Hippocampal cells were grown in this culture medium in 75 cm^3^ culture flasks at 37 °C and 5% CO_2_. On the ninth day, when astrocyte-enriched cell cultures reached 70–80% confluence, other cell types were eliminated through orbital rotation for a minimum of 16 h at 280 rpm and 37 °C. The cells were then harvested (0.05% trypsin/EDTA solution; Biochrom AG, Berlin, Germany), resuspended in DMEM: F12 (Gibco, Grand Island, NY, USA) supplemented with 10% fetal bovine serum (FBS) and 1% antibiotic-antimycotic solution, and centrifuged for 5 min at 233.6× *g*. Cells were then plated (P60-plate) at a density of 4.35 × 10^5^ cells/cm^2^ and cultured for 24 h. The medium was changed to serum-free media, and 24 h later, cells were collected. This procedure resulted in a purity of greater than 95% astrocytes. Finally, the following technical replicates from each experimental group were obtained: control 3×Tg-AD male: *n* = 9; ethanol 3×Tg-AD male: *n* = 9; control 3×Tg-AD female: *n* = 9; ethanol 3×Tg-AD female: *n* = 9.

### 4.5. RNA Isolation and Real-Time Quantitative PCR Analysis

We performed real-time PCR (TaqMan, ThermoFisher Scientific, Waltham, MA, USA) using specific sets of primer probes from TaqMan^®^ Gene Expression Assays as shown in Table 1. Total RNA was extracted from 60 mm plates of confluent astrocytes using the Trizol^®^ method according to the manufacturer’s instructions (ThermoFisher Scientific, Waltham, MA, USA). RNA samples were isolated with RNAeasy minelute cleanup-kit, including digestion with DNase I column (Qiagen, Hilden, Germany) and quantified using a spectrophotometer to ensure A260/280 ratios of 1.8–2.0. After the reverse transcript reaction from 1 μg of mRNA, a quantitative real-time reverse transcription-polymerase chain reaction (qPCR) was performed in a CFX96TM Real-Time PCR Detection System (Bio-Rad, Hercules, CA, USA) and the FAM dye-labeled format for the TaqMan^®^ Gene Expression Assays (ThermoFisher Scientific, Waltham, MA, USA). Each reaction was run in duplicate and contained 9 μL of cDNA diluted 1/100. Cycling parameters were: 50 °C for 2 min to deactivate single- and double-stranded DNA containing dUTPs, 95 °C for 10 min to activate Taq DNA polymerase, followed by 40 cycles at 95 °C for 15 s for cDNA melting, and 60 °C for 1 min to allow for annealing and the extension of the primers, during which fluorescence was acquired. A melting curve analysis was performed to ensure that only a single product was amplified. After analyzing several reference genes, values obtained from astrocytes were normalized in relation to Actb levels, which were found to be stable between experimental groups (*n* = 9/group). Relative quantification was performed using the ΔΔCt method with efficiency correction applied to each target-reference pair. Ct values with a standard deviation >0.5 in technical replicates were flagged for review. The first step was to calculate the normalized expression (ΔCt) for each sample. This was achieved by subtracting the average Ct value of the validated reference gene from the Ct value of the target gene for the same sample. Next, the relative expression (ΔΔCt) was calculated by subtracting the average ΔCt of the control group from the ΔCt of each individual experimental sample. Finally, the fold change in expression was calculated using the equation 2^−ΔΔCt^. Outliers were excluded if values deviated >2 SD from the mean and did not meet quality control thresholds.

### 4.6. Western Blot Analysis

Astrocyte cultures (six random replicates, *n* = 6/group) were prepared for Western blot analysis to measure protein levels. The cells were first homogenized in 500 μL of ice-cold lysis buffer. This buffer contained Triton X-100, 1 M4-(2-hydroxyethyl)-1-piperazineethanesulfonic acid (HEPES), 0.1 M ethylenediaminetetraacetic acid (EDTA), sodium pyrophosphate, sodium fluoride (NaF), sodium orthovanadate (NaOV), and protease inhibitors using a tissue-lyser system (Qiagen, Hilden, Germany). The resulting mixture was centrifuged for 30 min at 26,000× *g* at 4 °C to separate the soluble proteins, and the supernatant (containing the proteins) was collected. The total protein concentration in the samples was then determined using the Bradford method. A quantity of 30 μg of total protein from each sample was separated by size using 4–12% polyacrylamide gradient gels (SDS-PAGE). The separated proteins were then transferred from the gels onto nitrocellulose membranes for 1 h at 80 V in a buffer containing 20% methanol, 3.03% TRIS, and 14.4% Glycine. Successful transfer was confirmed by staining the membranes with Ponceau red. The membranes were first blocked for one hour at room temperature in a solution of TBS-T (Tris-Buffered Saline with 0.1% Tween 20) containing 2% BSA (bovine serum albumin; Roche, Mannheim, Germany) to prevent non-specific antibody binding. Next, membrans were incubated overnight (16–18 h) with appropriate primary antibodies diluted in a solution of TBS-T with 2% BSA. Table 2 contains a list of antibody references and dilutions used. After incubation, the membranes were extensively washed with TBS-T solution and then incubated for one hour at room temperature with a 1:10,000 dilution of HRP-conjugated anti-rabbit or anti-mouse IgG secondary antibody (anti-rabbit or anti-mouse IgG (H + L); Promega, Madison, WI, USA) in TBS-T with 2% BSA. Following extensive washing, the protein bands were visualized by incubating the membranes for one minute with the Western Blotting Luminol Reagent kit (Bio-Rad Laboratories, Inc., Hercules, CA, USA), and the resulting chemiluminescence was detected by using a ChemiDoc MP Imaging System (Bio-Rad Laboratories, Inc., Hercules, CA, USA). Mouse β-actin was included as a reference protein. Finally, the protein levels were quantified with ImageJ software (Version 1.54p). Densitometric values were normalized to those of reference protein, loading controls and consistent background subtraction. Band intensities were verified within the validated linear range of optimal loading amount for accurate detection, which was previously assessed in previous studies run in the laboratory [47,48].

### 4.7. Endocannabinoid and Paracannabinoid Quantification

Culture medium samples (five random replicates, *n* = 5/group) were analyzed following the method indicated in [16,49]. For the preparation of the samples, 500 µL of volume was used, and 2 biological replicates per sample were made, each analyzed in triplicate (technical replicates), in an HPLC-MS. The chromatographic separation was carried out in an HPLC model Ultimate 3000 from Thermo Scientific (Waltham, MA, USA). The column was ACE Excel 2 C18 (2 mm particle size, 10 × 3.0 mm ACE) maintained at 40 °C with a mobile phase flow rate of 0.3 mL/min. The composition of the mobile phase was: A, 0.1% (*v*/*v*) formic acid in water, and B, 0.1% (*v*/*v*) formic acid and 0.05 mM of NaCl in acetonitrile. The initial conditions were 60% A and 40% B. The gradient was increased linearly to 100% B over 5 min, maintained at 100% B for 3 min, and returned to the initial conditions for a further 4 min with a total run time of 20 min. The mass spectrometer was a Q Exactive from Thermo Scientific with an Orbitrap detector. The spectral range was recorded between 270 and 410 m/z in positive and negative mode for: POEA, LEA, 2AG, 2LG, PEA, OEA, and SEA. The desolvation gas temperature was 230 °C, a gas flow rate of 40 mL/min, auxiliary gas 15 mL/min, and sweep gas 2 mL/min. The spray voltage was set at 3.0 kV. The compounds were identified by their exact mass with an exactitude of less than 2 ppm. A six-point external calibration curve was prepared with concentrations ranging from 0.5 to 15 ng/mL for SEA, LEA, and OEA, and from 10 to 300 ng/mL for 2AG, PEA, and POEA. Internal standards included SEA-d_4_, LEA-d_4_, PEA-d_4_, and OEA-d_4_ at 1.5 ng/mL, and 2AG-d_5_ at 15 ng/mL.

### 4.8. Statistical Analysis

Statistical analyses were performed with IBM SPSS Statistics 23 and GraphPad Prism 9. Data are represented as the mean ± standard error of the mean (SEM). An initial sample size (*n*) of 9, 6, and 5 per experimental group was used as technical replicates. Outliers were identified following the ROUT method (Q = 5%). See Appendix A for further information. Before conducting the two-way analysis of variance (two-way ANOVA), the underlying assumptions were evaluated to ensure the validity of the results. First, the assumption of independence of observations was met through an appropriate study design and random assignment. Second, the normality of residuals was assessed using visual inspection of Q-Q plots and confirmed with the Shapiro–Wilk test, indicating no significant deviations from normality. Third, homogeneity of variances across groups was tested using Levene’s test, which showed that the variances were sufficiently equal (*p* > 0.05). These checks support the use of two-way ANOVA for analyzing the data, whose factors were sex (males and females) and perinatal exposure (water and ethanol). Post hoc multiple comparisons were conducted using Tukey’s test. Effect sizes can be found in the Appendix A. Sample size, outliers and degrees of freedom can be found in Appendix A. A *p*-value of less than 0.05 was considered statistically significant.

## 5. Conclusions

This study provides novel insights into GPR55/PEA-mediated signaling in hippocampal astrocytes following PEE, emphasizing its potential role in shaping early neurobiological vulnerability to AD. GPR55, a G protein-coupled receptor, and PEA, a lipid-derived mediator, are key regulators of neuroprotection and inflammation, particularly through astrocytic pathways. Our findings indicate that PEE induces a pronounced astrocytic response characterized by elevated neuroinflammation and diminished GPR55/PEA signaling, especially in male offspring, suggesting sex-dependent mechanisms of increased susceptibility to AD-related pathology. These results underscore the importance of astrocytic endocannabinoid signaling in the developmental origins of AD and support further investigation using PEE-exposed 3×Tg-AD mice in longitudinal designs on GPR55/PEA as a potential therapeutic target to guide future strategies aimed at mitigating the long-term effects of prenatal insults on brain health.

## Figures and Tables

**Figure 1 ijms-26-11154-f001:**
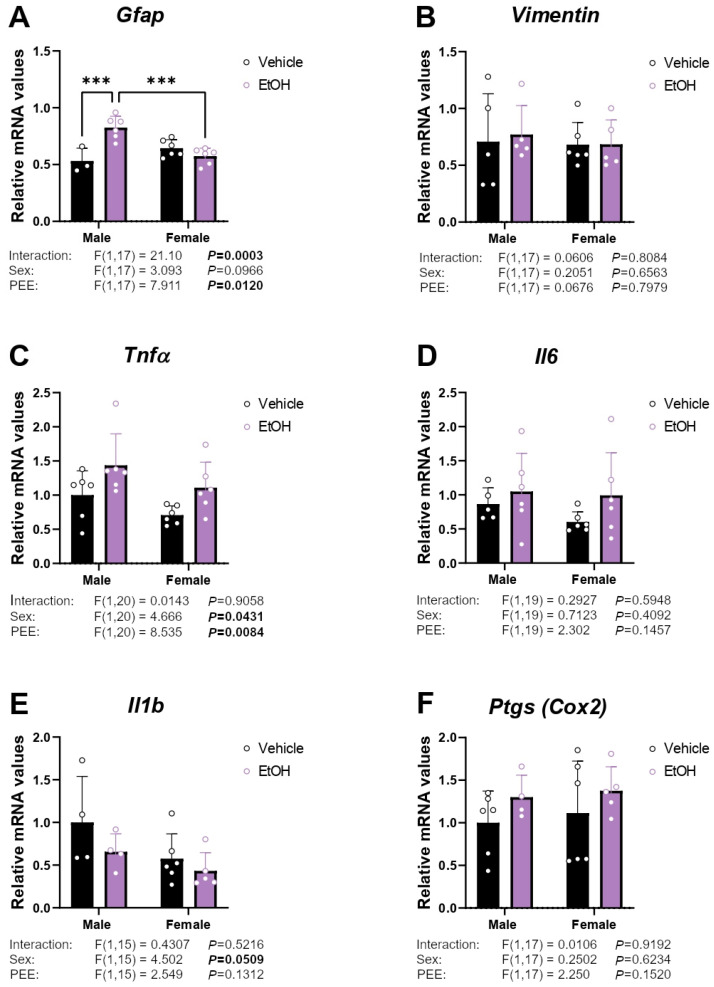
mRNA expression of the inflammation factors *Gfap* (**A**), *Vimentin* (**B**), *Tnfα* (**C**), *Il6* (**D**), *Il1β* (**E**), and *Ptgs* (**F**) in hippocampal astrocytes from 3×Tg-AD offspring of both sexes born to control and PEE mothers. Data are expressed as the mean ± S.E.M. (*n* = 4–6). Relevant *p* values from the two-way ANOVA are shown below each graph, with significant *p*-values highlighted in bold. Tukey’s test: *** *p* < 0.001. See also effect sizes in the Appendix A.

**Figure 2 ijms-26-11154-f002:**
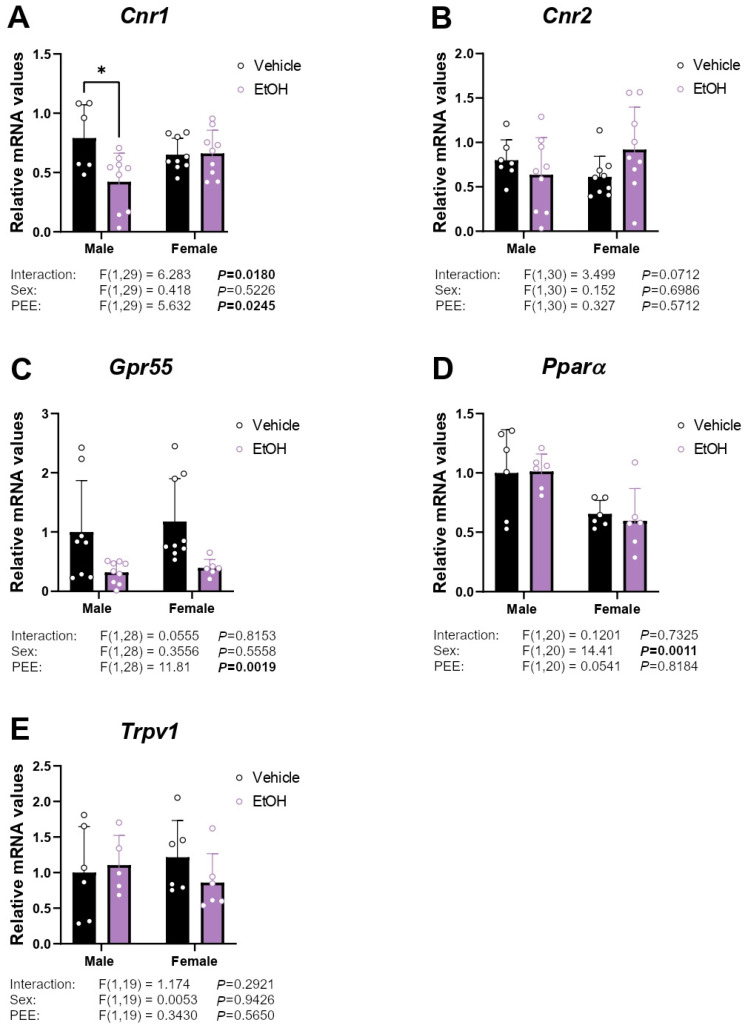
mRNA expression of the cannabinoid-related receptors *Cnr1* (**A**), *Cnr2* (**B**), *Gpr55* (**C**), *Pparα* (**D**), and *Trpv1* (**E**) in hippocampal astrocytes from 3×Tg-AD offspring of both sexes born to control and PEE mothers. Data are expressed as the mean ± S.E.M. (*n* = 5–9). Relevant *p* values from the two-way ANOVA are shown below each graph, with significant *p*-values highlighted in bold. Tukey’s test: * *p* < 0.05. See also effect sizes in the Appendix A.

**Figure 3 ijms-26-11154-f003:**
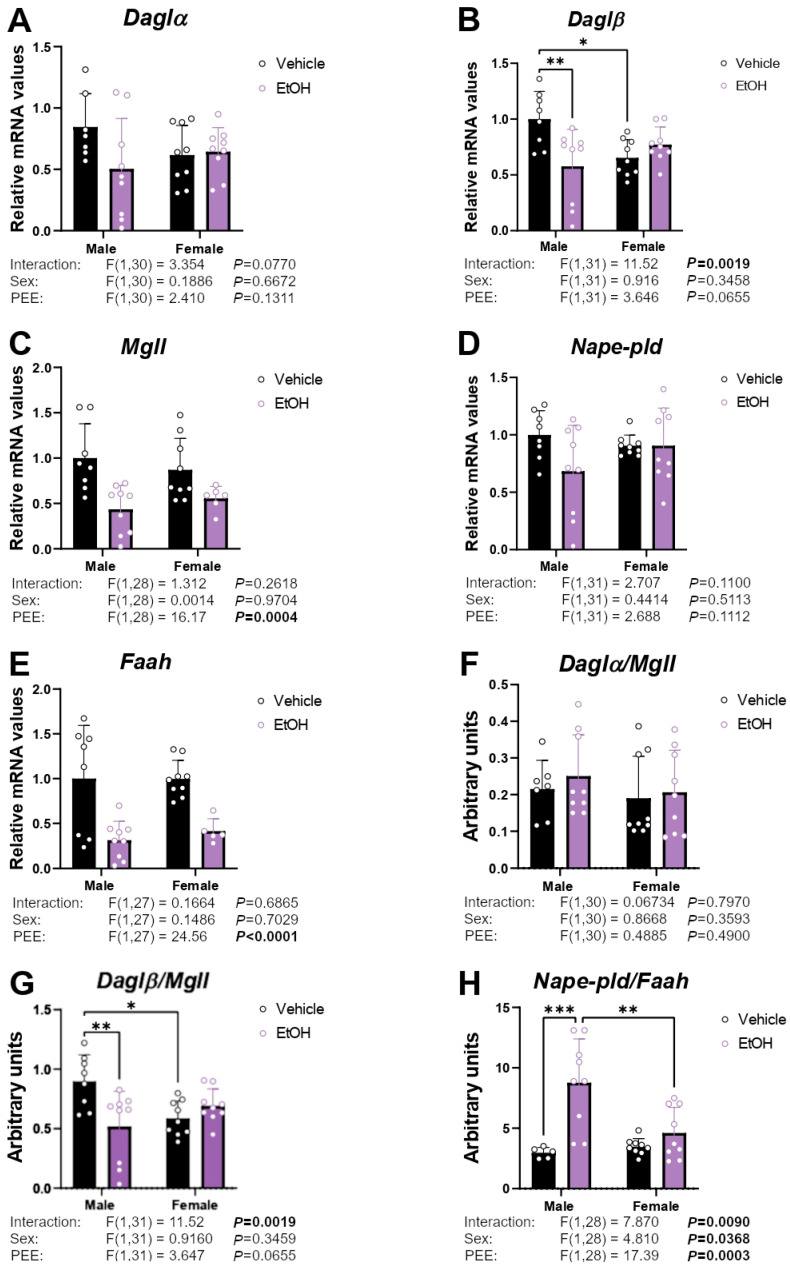
mRNA expression of the endocannabinoid enzymes *Daglα* (**A**), *Daglβ* (**B**), *Mgll* (**C**), *Nape-pld* (**D**), and *Faah* (**E**), and the ratios *Daglα*/*Mgll* (**F**), *Daglβ*/*Mgll* (**G**), and *Nape-pld*/*Faah* (**H**) in hippocampal astrocytes from 3×Tg-AD offspring of both sexes born to control and PEE mothers. Data are expressed as the mean ± S.E.M. (*n* = 5–9). Relevant *p* values from the two-way ANOVA are shown below each graph, with significant *p*-values highlighted in bold. Tukey’s test: */**/*** *p* < 0.05/0.01/0.001. See also effect sizes in the Appendix A.

**Figure 4 ijms-26-11154-f004:**
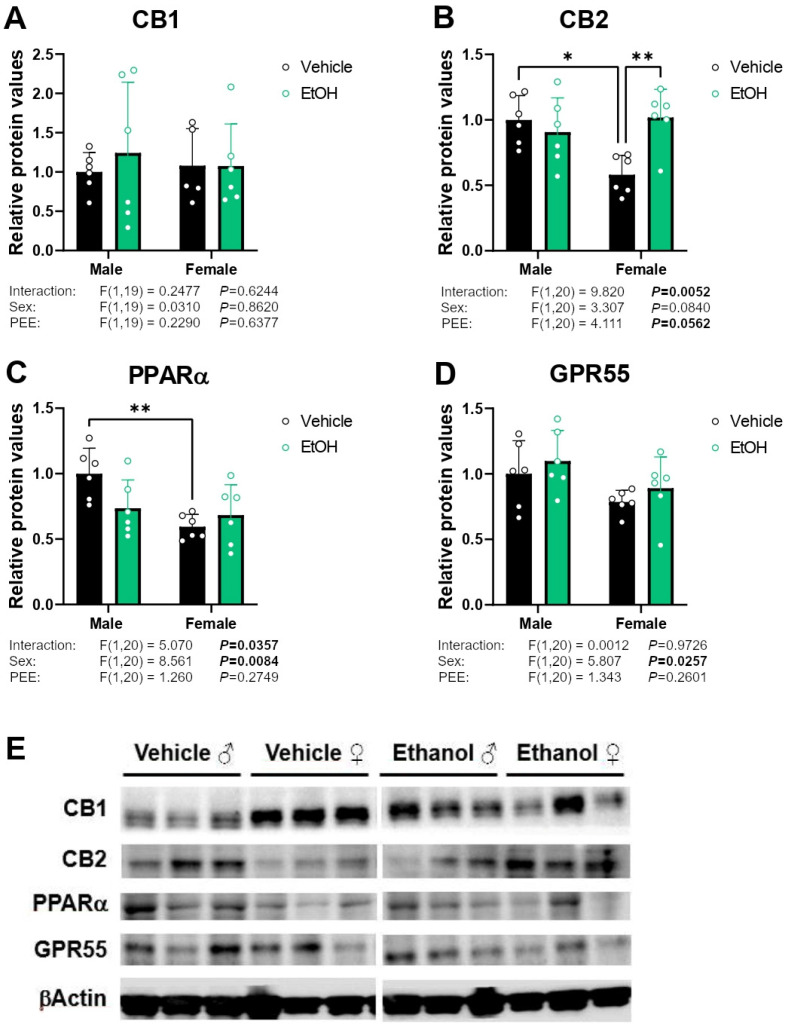
Protein expression of the cannabinoid-related receptors CB1 (**A**), CB2 (**B**), PPARα (**C**), and GPR55 (**D**) in hippocampal astrocytes from 3×Tg-AD offspring of both sexes born to control and PPE mothers. Representative immunoblots (**E**). Data are expressed as the mean ± S.E.M. (*n* = 5–6). Relevant *p* values from the two-way ANOVA are shown below each graph, with significant *p*-values highlighted in bold. Tukey’s test: */** *p* < 0.05/0.01. See also effect sizes in the Appendix A. See unedited blots in the Appendix A.

**Figure 5 ijms-26-11154-f005:**
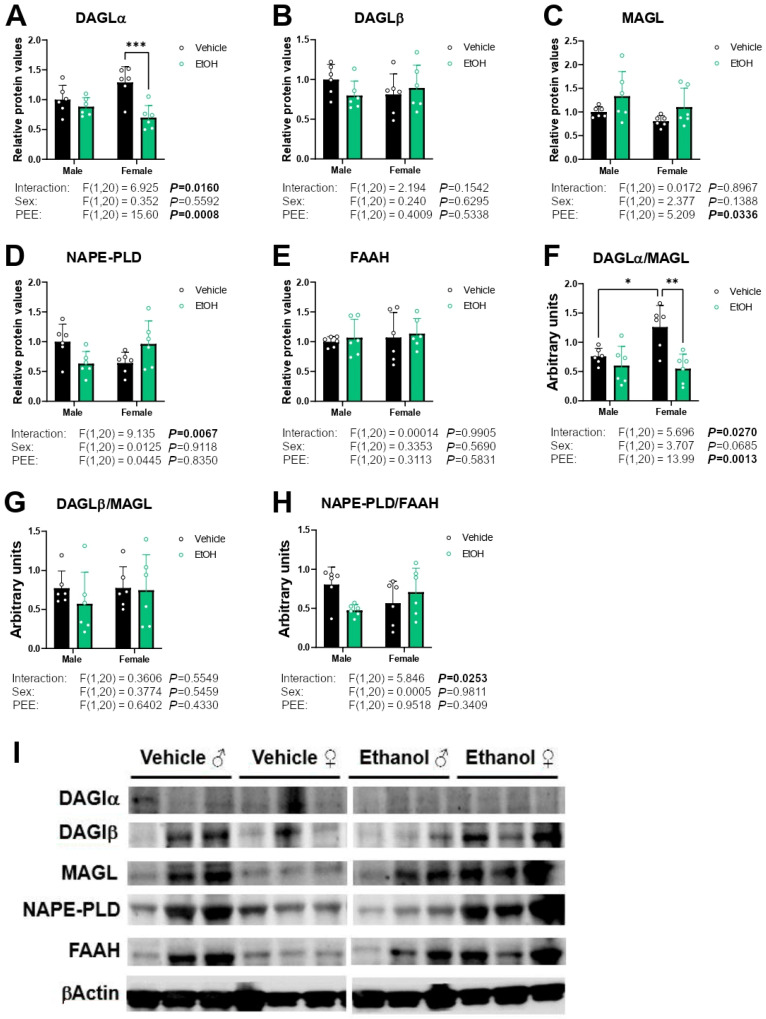
Protein expression of the endocannabinoid enzymes DAGLα (**A**), DAGLβ (**B**), MAGL (**C**), NAPE-PLD (**D**), and FAAH (**E**), and the ratios DAGLα/MAGL (**F**), DAGLβ/MAGL (**G**), and NAPE-PLD/FAAH (**H**) in hippocampal astrocytes from 3×Tg-AD offspring of both sexes born to control and PPE mothers. Representative immunoblots (**I**). Data are expressed as the mean ± S.E.M. (*n* = 6). Relevant *p* values from the two-way ANOVA are shown below each graph, with significant *p*-values highlighted in bold. Tukey’s test: */**/*** *p* < 0.05/0.01/0.001. See also effect sizes in the Appendix A. See unedited blots in the Appendix A.

**Figure 6 ijms-26-11154-f006:**
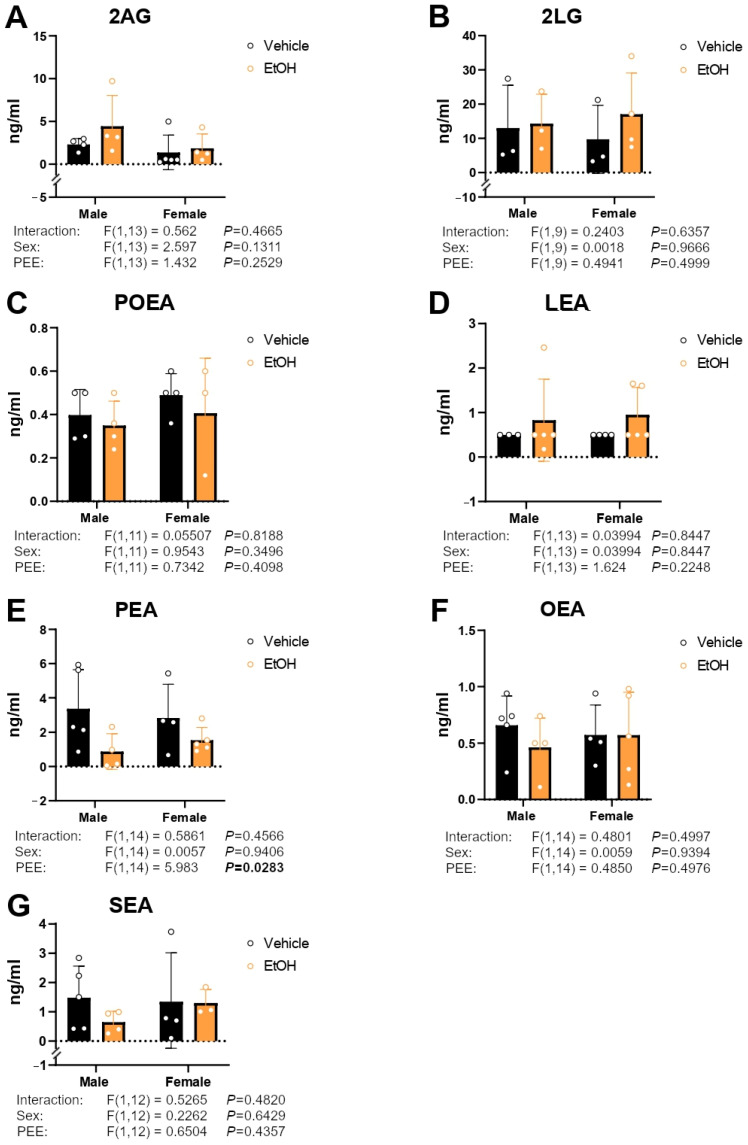
Concentrations of the acylglycerols 2AG (**A**) and 2LG (**B**), and the NAEs POEA (**C**), LEA (**D**), PEA (**E**), OEA (**F**), and SEA (**G**) in the culture medium of hippocampal astrocytes from 3×Tg-AD offspring of both sexes born to control and PEE mothers. Data are expressed as the mean ± S.E.M. (*n* = 3–5). Relevant *p* values from the two-way ANOVA are shown below each graph, with significant *p*-values highlighted in bold. See also effect sizes in the Appendix A.

**Figure 7 ijms-26-11154-f007:**
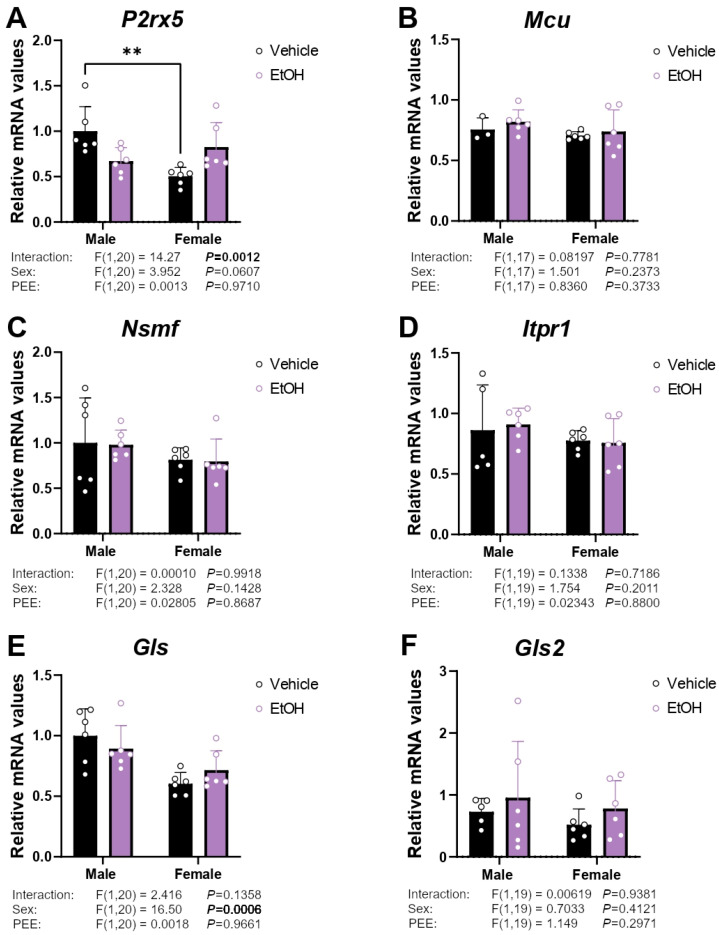
mRNA expression of the Ca^2+^ signaling factors *P2rx5* (**A**), *Mcu* (**B**), *Nsmf* (**C**), *Itpr1* (**D**), *Gls* (**E**), and *Gls2* (**F**) in the hippocampal astrocytes from 3×Tg-AD offspring of both sexes born to control and PEE mothers. Data are expressed as the mean ± S.E.M. (*n* = 3–6). Relevant *p* values from the two-way ANOVA are shown below each graph, with significant *p*-values highlighted in bold. Tukey or single-effect analysis: ** *p* < 0.01. See also effect sizes in the Appendix A.

**Figure 8 ijms-26-11154-f008:**
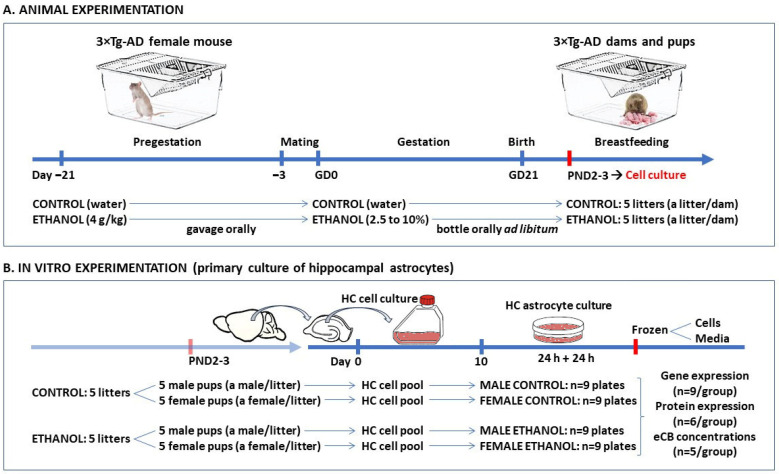
Schematic representation following the temporal design of the animal (**A**) and in vitro experimentation (**B**). Experimental procedures on 3×Tg-AD female mice were carried out from a pregestational period of 21 days before mating until postnatal day (PND) 2–3 of the offspring. Finally, 5 litters born to control mothers and 5 litters born to prenatal ethanol-exposure (PEE) mothers were randomly selected (a litter per dam). The in vitro procedures for primary hippocampal astrocyte culture began with the random selection of 5 male and 5 female offspring (a pup of each sex per litter) from the control and PEE litters (4 experimental groups). Following brain extraction and dissection, and homogenization of the hippocampus (HC), a pool of hippocampal cells from 5 pups of each sex from each condition was cultured for 10 days. The astrocyte-enriched HC cell cultures were divided into 9 plates from each experimental group (*n* = 9/group). Cells and culture media were frozen for later analysis.

**Table 1 ijms-26-11154-t001:** Primer references for Taqman^®^ Gene Expression Assays.

Gene Symbol	Assay ID	GenBank Accession Number	Amplicon Length (bp)
*Actb*	Mm02619580	NM_007393.5	143
*Gfap*	Mm01253033	NM_001131020.1	75
*Vim (Vimentin)*	Mm01333430	NM_011701.4	62
*Tnfa*	Mm00443258	NM_001278601.1	81
*Il1b*	Mm00434228	NM_008361.3	90
*Gls*	Mm01257297	NM_001081081.2	114
*Gls2*	Mm01164862	NM_001033264.3	118
*Cnr1 (CB1)*	Mm01212171	NM_007726.3	66
*Cnr2 (CB2)*	Mm02620087	NM_009924.4	171
*Gpr55*	Mm02621622	NM_001033290.2	102
*Ppara*	Mm00440939	NM_001113418.1	74
*Trpv1*	Mm01246300	NM_001001445.2	56
*Dagla*	Mm00813830	NM_198114.2	69
*Daglb*	Mm00523381	NM_144915.3	72
*Nape-pld*	Mm00724596	NM_178728.5	85
*Mgll*	Mm00449274	NM_001166249.1	78
*Faah*	Mm00515684	NM_010173.4	62
*P2rx5*	Mm00473677	NM_033321.3	104
*Mcu*	Mm01168773	NM_001033259.4	71
*Nsmf*	Mm00480341	NM_001039386.1	70
*Itpr1*	Mm00439907	NM_010585.5	58
*Il6*	Mm00446190	NM_031168.1	78
*Ptgs2*	Mm00478374	NM_011198.3	80

**Table 2 ijms-26-11154-t002:** Antibodies used for protein expression by Western blotting.

Antigen	Immunogen	Manufacturing	Dilution
β-actin	A purified Arabidopsis actin protein.	Abcam (Cambridge, UK). (#ab230169) Mouse monoclonal antibody	1:2000
CB1	Synthetic Peptide within Human CNR1 aa 450 to C-terminus.	Abcam (Cambridge, UK). (#ab23703) Rabbit polyclonal antibody	1:200
CB2	Recombinant Fragment Protein within Rat Cnr2 aa 1–50.	Abcam (Cambridge, UK). (#ab3561) Rabbit polyclonal antibody	1:200
PPARα	Synthetic Peptide within Human PPARA aa 300–400 conjugated to KLH.	Abcam (Cambridge, UK). (#ab215270) Rabbit polyclonal antibody	1:500
GPR55	Synthetic peptide from an internal cytoplasmic region of human GPR55	Cayman (Ann Arbor, MI, USA). #10224 Rabbit polyclonal antibody	1:200
DAGLα	Synthetic peptide from the central region of human DAGLA (317–345 aa).	Biorbyt (Cambridge, UK) #orb1561573 Rabbit polyclonal antibody	1:100
DAGLβ	KLH conjugated synthetic peptide derived from human DAGLB (51–150/672aa)	Biorbyt (Cambridge, UK) #orb182976 Rabbit polyclonal antibody	1:200
NAPE-PLD	Synthetic Peptide within Human NAPEPLD aa 150–200.	Abcam (Cambridge, UK). #ab95397 Rabbit polyclonal antibody	1:200
MAGL	Recombinant Fragment Protein within Human MGLL aa 1–50.	Abcam (Cambridge, UK). #ab24701 Rabbit polyclonal antibody	1:200
FAAH	Synthetic peptide from the C-terminal region of rat FAAH	Cayman (Ann Arbor, MI, USA). #101600 Rabbit polyclonal antibody	1:200

## Data Availability

The data that support the findings of this study are available from the corresponding author upon reasonable request.

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
