# Peer review of "Perinatal Ethanol Exposure Induces Astrogliosis and Decreases GRP55/PEA-Mediated Neuroprotection in Hippocampal Astrocytes of the 3×Tg Alzheimer’s Animal Model"

_ijms, 2025, doi:10.3390/ijms262211154_

Round 1
Reviewer 1 Report
Comments and Suggestions for Authors
The first question for the authors is why an animal model was necessary. Did the authors initially examine the effects on normal human astrocytes by exposing them to alochola and all the marker genes mentioned in the current study? Alternatively, did they compare normal astrocytes with AD-affected astrocytes in the molecular studies they performed?
Most of the experimental sections are not well explained such as RNA isolation procedures and how the samples were prepared for 'endocannabinoid and p-cannabinoid quantification.' Overall, the manuscript is not well structured, making it difficult to differentiate between the results and understand what exactly was done in the experimental section.
Additionally, why did the authors choose to perform targeted analysis in the mass spectrometry rather than an untargeted approach for a more comprehensive comparison? The results from the mass spectrometry are also not clearly explained.
Reviewer 2 Report
Comments and Suggestions for Authors
Manuscript Title:
Perinatal ethanol exposure induces astrogliosis and decreases GRP55/PEA-mediated neuroprotection in hippocampal astrocytes of the 3×Tg Alzheimer's animal model
Summary
This paper explores how perinatal ethanol exposure (PEE) affects astrocytic endocannabinoid and inflammatory signalling in a 3×Tg-AD mouse model. The authors looked at molecular markers like GFAP, TNFα, CB1, GPR55, and PEA signalling in primary astrocytes from neonatal hippocampi. Overall, it’s an interesting study with a novel focus on GPR55/PEA signalling and sex differences. The results suggest potential mechanisms linking early ethanol exposure to Alzheimer's-related vulnerability, which I think is an important question. The manuscript is generally well-organized, but some areas could be clearer and more cautiously interpreted.
General Comments
- The study is relevant and addresses a question that hasn’t been widely explored. I like the focus on sex differences and the breadth of molecular analyses.
- Some interpretations in the Discussion feel stronger than the data allow; it might be better to frame them as possibilities rather than definitive conclusions.
- The ethical approval is clearly stated, which is good. However, more details about whether replicates are biological or technical would help with reproducibility.
Section-Specific Comments
Abstract
- The abstract is clear overall, but a few sentences are quite long and could be split up.
- Methods are only mentioned briefly; it would help to state that primary astrocytes were isolated from neonatal hippocampi.
- Words like “increased” or “decreased” are a bit vague, adding “significant” or approximate fold-changes would make it stronger.
- The last sentence about AD risk feels a bit overreaching; I’d suggest softening it to highlight potential mechanisms rather than direct disease predisposition.
Introduction
- The background is strong and explains the knowledge gap well.
- Some references are older; including more recent studies (2020–2023) would strengthen the novelty.
- There’s a paragraph on 2-AG that appears twice; this could be combined.
- The study's aim could be framed as a clear hypothesis:
“We hypothesize that PEE alters astrocytic endocannabinoid and inflammatory signalling, which might increase susceptibility to AD pathology later in life.”
Materials and Methods
- Overall, well-organised and detailed; ethical approval is stated.
- It would help to clarify whether replicates are biological or technical.
- Could provide more detail on ethanol dosing rationale, antibody validation, and HPLC-MS quality control.
- Reporting total animal numbers (dams, litters, offspring) would be useful.
- Mentioning whether ANOVA assumptions were checked and including effect sizes would improve transparency.
- A note about the limitations of extrapolating from in vitro astrocytes to in vivo systems would help balance the interpretation.
Results
- Logical structure, moving from inflammatory markers to receptor expression, lipids, and calcium signalling.
- Sample sizes (n) should be clearly stated in figures or text.
- Figures are mostly clear, but adding exact p-values and clarifying comparisons would help.
- Sex differences are interesting, but it’s not always clear whether this was exploratory or predefined.
- Highlighting key results in a summary figure or schematic could help readers.
- Some interpretive statements (e.g., “probably due to deregulated enzymatic machinery”) would be better in the Discussion.
- Clarifying replication (independent litters vs. repeated wells) would improve confidence.
- Some functional relevance of sex-specific gene changes (like P2rx5, Gls) could be briefly described.
Discussion
- Integrates results well, especially the focus on GPR55/PEA signalling.
- Some claims are phrased too strongly; I’d suggest presenting them as hypotheses or possibilities.
- Speculative ideas (like FAAH/NAPE-PLD compensatory changes) should be clearly noted as such.
- Updating references with more recent studies (2020–2023) on astrocytes, ECS, and excitotoxicity would make it more current.
- The discussion of PEA’s therapeutic potential is long; condensing it and focusing on findings relevant to this study could improve readability.
- Sex differences are interesting but could be expanded; why male astrocytes seem more vulnerable, and whether this aligns with FASD or AD epidemiology, could be discussed.
- Limitations could be more fully addressed, including the in vitro nature of experiments, lack of functional assays, and null results.
Conclusions
- Summarises the main findings clearly: reduced GPR55/PEA signalling after PEE.
- Avoid overstating translational implications; softer phrasing like “may contribute to” or “suggests a potential role in” would be better.
- Could briefly note limitations and point toward future in vivo or translational work.
References
- Overall, broad and relevant, including some recent studies (2021–2023).
- It could include more recent work on astrocytes, neuroinflammation, and sex differences.
- Balance PEA literature by including studies showing limitations or inconsistencies.
- Include more human or translational studies.
- Some formatting issues: Ref. 14 merges two citations; Ref. 36 merges two; Ref. 24 lacks a page number.
- Consider removing references that are not directly relevant (e.g., basal ganglia, cocaine sensitization).
Specific Comments
- Figures: include exact p-values and clarify comparisons.
- Results: report null findings to be transparent.
- Methods: clarify biological vs. technical replicates.
- Discussion: tone down mechanistic claims that aren’t directly supported.
- References: fix formatting, update literature, balance PEA evidence.
Overall Assessment
This is an interesting and well-thought-out study with a novel focus on GPR55/PEA signalling and sex differences. The experimental design is generally sound, and the results are presented clearly. That said, the manuscript would benefit from:
- Clearer presentation of some results.
- Softer interpretation of mechanistic claims.
- Expanded discussion of limitations.
- Updating references with more recent studies.
- Minor formatting fixes.
The quality of the English language is good.
Reviewer 3 Report
Comments and Suggestions for Authors
Comments
Results
My feedback on this section will be limited because it is unclear whether pups were treated as independent experimental units (which could invalidate comparisons), and there are also statistical reporting issues that restrict the depth of my evaluation. The comparison is not clear: in the Methods they state that all analyses were performed using two-way ANOVA (sex and treatment as factors), but here they do not present the full ANOVA model in a systematic way. Instead, they highlight only one or two effects and then move directly to post hoc testing. The correct procedure is to first report the complete two-way ANOVA, showing F values, degrees of freedom, and p-values for each term (sex, treatment, and interaction). Only then, if an interaction is significant, post hoc comparisons between groups are justified. If only a main effect of treatment is present, then the statement should be restricted to “offspring of ethanol-exposed dams show alterations in X,” but not more. In addition, the n per group should be explicitly indicated to assess whether any exclusions were made. As it is currently written, the reporting is not appropriate.
Materials and Methods
4.2 Animal Model
Report the exact ages (in weeks) of females at habituation, mating and gestation; the initial N of dams per group and pregnancy rates; whether pregnant dams were singly housed; and the actual ethanol intake per dam (mL/day and g/kg/day) or blood ethanol levels. Provide estimated gestational dose (mean±SD per week), availability of plain water, and reproductive outcomes (litter size, neonatal survival, sex ratio, PND1 body weight). Clarify randomization/blinding. Given litter-based outcomes, please state explicitly that the litter is treated as the experimental unit.
4.3 Primary astrocytes
The current description lists n=9 per sex/condition but does not specify how many litters contributed to those samples. Because pups from the same litter are not independent observations, the litter—not the pup—must be considered the experimental unit for offspring-derived measures. Please report the number of litters per group, the number of pups sampled per litter, and the analytic strategy to avoid pseudoreplication (e.g., one culture per litter, litter-averaging prior to analysis, or mixed-effects models with litter as a random factor).
4.4 RNA isolation and RT-qPCR
Provide MIQE-compliant details: reference gene validation (preferably ≥2 stable genes), per-amplicon PCR efficiencies and r², quantification method (ΔΔCt with efficiency correction), number of biological vs technical replicates, QC criteria for Ct dispersion/outliers, and whether housekeeping expression was unaffected by PEE.
4.5 Western blot analysis
Add transfer and blocking conditions (time/temperature), primary/secondary incubation details, linear-range validation of loaded protein.
4.6 Endocannabinoid / paracannabinoid quantification (HPLC-MS)
Expand sample prep and LC-MS QA/QC: extraction/storage conditions, randomized run order with pooled QC/blank injections, calibration model (range, r², weighting), LOD/LOQ, recovery and matrix-effect assessment, injection volume and autosampler stability, and acceptance criteria (e.g., CV%).
Summary: Because multiple pups from the same dam are not independent, analyses based on offspring-derived astrocyte cultures should treat the litter as the experimental unit (or include litter as a random effect in mixed models). Please report the number of litters per group and adjust the analysis accordingly.
Abstract, Introduction and Discussion
Given these statistical and reporting issues, I cannot provide a proper evaluation of the Introduction and Discussion at this stage. Without results that are analyzed and presented correctly, any comments on how the findings are framed or interpreted would be premature. In other words, a deeper review of these sections is not possible until the Results are made statistically sound and transparent.
Comments on the Quality of English LanguageThe overall quality of English in the manuscript requires improvement. Several sections contain grammatical errors, typographical mistakes, and inconsistencies in terminology that affect clarity. I recommend that the authors have the text carefully revised and edited by a fluent English speaker, preferably a native, before resubmission.
Round 2
Reviewer 1 Report
Comments and Suggestions for Authors
All the questions have been addressed. Accept in its current form.
Author Response
Thank you
Reviewer 3 Report
Comments and Suggestions for Authors
Comments
Given the current statistical and reporting inconsistencies, I am unable to provide a meaningful evaluation of the Abstract, Introduction, Discussion, and Conclusion. Until the results are properly analyzed and transparently presented, any interpretation of how the findings are framed or discussed would be speculative. A thorough assessment of these sections will only be possible once the Results are statistically sound and clearly reported.
Results
The statistical reporting and approach remain inconsistent. The F values from the two-way ANOVA must be reported in the text, not only within figure panels; this applies to all Results subsections, as statistical outputs are part of the data presentation and should be integrated into the narrative. The degrees of freedom vary across figures (e.g., F(1,17), F(1,20), F(1,22)), which is inconsistent with the stated 2×2 design (sex × treatment) and n = 9 per group; the authors must explain these discrepancies and clarify whether the effective sample size differed among analyses or whether exclusions occurred (some figure legends list larger n than in Methods), and explicitly report any exclusions. According to the Methods, Fisher’s LSD was applied when no interaction was found; this is not appropriate for a 2×2 design, where no post hoc test is required in the absence of interaction, and selective use of LSD inflates type I error. Moreover, in the cover letter, the authors justify this approach by citing Wei et al. (2012, Amino Acids, doi:10.1007/s00726-011-0924-0), claiming that the use of an uncorrected Fisher’s LSD test is acceptable when there is a main effect but no interaction. This justification is incorrect: the cited paper does not support this practice but rather discusses specific, hypothesis-driven comparisons under limited conditions. The selective use of uncorrected Fisher’s LSD in this context remains statistically unjustified and confirms a conceptual misunderstanding of the two-factorial design. These issues should be corrected to ensure statistical transparency and internal consistency.
Materials and Methods
In Section 4.1 (Animals), the authors do not specify whether the males used for mating were adults or sexually active, nor whether females were housed individually during pregnancy. There is also no information on whether ethanol exposure affected fertility, litter size, sex ratio, or neonatal survival. These details are essential to evaluate potential prenatal effects and to confirm that breeding conditions were consistent across groups. In Section 4.2 (Prenatal Ethanol Exposure), it is unclear whether liquid consumption was monitored, which is critical to verify exposure consistency among dams. The authors should indicate whether the solution was the sole fluid source or offered in parallel with water, and report any changes in drinking behavior or pregnancy outcomes associated with ethanol intake. In Section 4.6 (HPLC–MS), the paragraph remains redundant and unclear. The first and last sentences repeat the same information regarding internal standards. Reported units are inconsistent: “500 mL” is implausibly large for this analysis (likely 500 µL), and “ng/L” should probably read “ng/mL.” Commas are used instead of decimal points in numerical values. The statement “2 replicates measured three times each” is ambiguous, as it is not specified whether these are biological or technical replicates.
Parameters showing no significant differences could be omitted from the figures and briefly described in the text, stating that no differences were found and providing the corresponding statistical values. This would help declutter the figures, which are currently overloaded with graphs that add little informative value.
Comments on the Quality of English LanguageThe English language has improved compared to the previous version, showing better grammar and overall clarity. However, the text still contains awkward phrasing, redundancies, and non-idiomatic constructions that affect fluency. A thorough revision by a native or fluent English editor is recommended to ensure linguistic accuracy and readability.
Round 3
Reviewer 3 Report
Comments and Suggestions for Authors
Comments
The main statistical issues identified in previous rounds remain unresolved. The reporting of two-way ANOVA results continues to be inconsistent and incomplete, with varying degrees of freedom across analyses that are incompatible with the declared group sizes. The authors’ previous explanation—attributing these discrepancies to cDNA inconsistencies and outlier removal—does not adequately justify the magnitude or frequency of the variation observed. This raises concerns about whether all analyses were conducted with the same dataset structure or if technical replicates were treated as independent observations, which would constitute pseudoreplication.
In several sections, only one F value is reported per analysis, often ambiguously linked to multiple effects (main and interaction), which is statistically incorrect. Each factor and the interaction must be reported with its own F, degrees of freedom, and p-value, regardless of significance. Without complete and transparent statistical reporting, the validity of the results cannot be properly evaluated.
Additionally, inconsistencies between the representative Western blot images and their corresponding quantitative graphs remain evident. In some cases, the blots do not visually support the magnitude or direction of the reported effects, which undermines data reliability.
Overall, these recurring problems indicate a lack of statistical transparency and analytical rigor that still prevents a proper interpretation of the study’s findings.
Comments on the Quality of English LanguageThe grammar is improved, but the text still reads awkwardly in several parts, with non-idiomatic phrasing and redundancies that affect fluency. Automated tools like Grammarly are not sufficient for publication quality; a final revision by a professional English editor is still recommended.
